# STViT: Improving Self-supervised Multi-Camera Depth Estimation with Spatial-Temporal Context and Adversarial Geometry Regularization

## Abstract

Multi-camera depth estimation has recently garnered significant attention due to its practical implications in autonomous driving. While adapting monocular self-supervised methods to the multi-camera context has demonstrated promise, these techniques often overlook unique challenges specific to multi-camera setups, hindering the realization of their full potential. In this paper, we delve into the task of self-supervised multi-camera depth estimation and propose an innovative Transformer-based framework, STViT, featuring several noteworthy enhancements: 1) The Spatial-Temporal Transformer (STTrans) is designed to exploit local spatial connectivity and global context within image features, facilitating the learning of enriched spatial-temporal cross-view correlations and effectively recovering intricate 3D geometries. 2) To alleviate the adverse impact of varying illumination conditions in photometric loss calculation, we employ a spatial-temporal photometric consistency correction strategy (STPCC) to adjust the image intensities and maintain brightness consistency across frames. 3) In recognition of the profound impact of adverse conditions such as rainy weather and nighttime driving on depth estimation, we propose an Adversarial Geometry Regularization (AGR) module based on Generative Adversarial Networks. The AGR serves to provide added spatial positional constraints on depth estimation by leveraging unpaired normal-condition depth maps, effectively preventing improper model training in adverse conditions. Our approach is extensively evaluated on large-scale autonomous driving datasets, including Nuscenes and DDAD, demonstrating its superior performance, thus advancing the state-of-the-art in multi-camera self-supervised depth estimation.

## 1 Introduction

Depth estimation involves assigning a depth value to each pixel for input RGB images, indicating the distance of the corresponding 3D point from the camera. This task constitutes a foundational aspect of perceiving the 3D geometric structure of the environment. It serves as a fundamental technology underpinning critical applications such as autonomous driving, robotics, drones, VR/AR, etc.

With the advent of the deep learning techniques (Krizhevsky et al., 2012; He et al., 2016), supervised depth estimation has gained significant attention. These approaches primarily involve the use of high-precision devices like LiDAR to generate ground truth depth from 3D point clouds, which are then used to supervise network training. Typically treating depth estimation as a regression (Eigen et al., 2014; Yin et al., 2019) or classification (Fu et al., 2018; Bhat et al., 2021) problem, these methods have exhibited impressive performance, thereby propelling advancements in the realm of 3D perception. However, due to the difficulty and high cost of obtaining LiDAR devices, accurate depth ground truth is rarely available in practical applications, which impedes the feasibility of supervised depth estimation methods.

Consequently, numerous successful endeavors have emerged in the realm of self-supervised depth estimation. These methods leverage photometric consistency across consecutive frames as a supervisory signal, concurrently optimizing depth and pose estimation. In the general pipeline, a depth network and a pose network

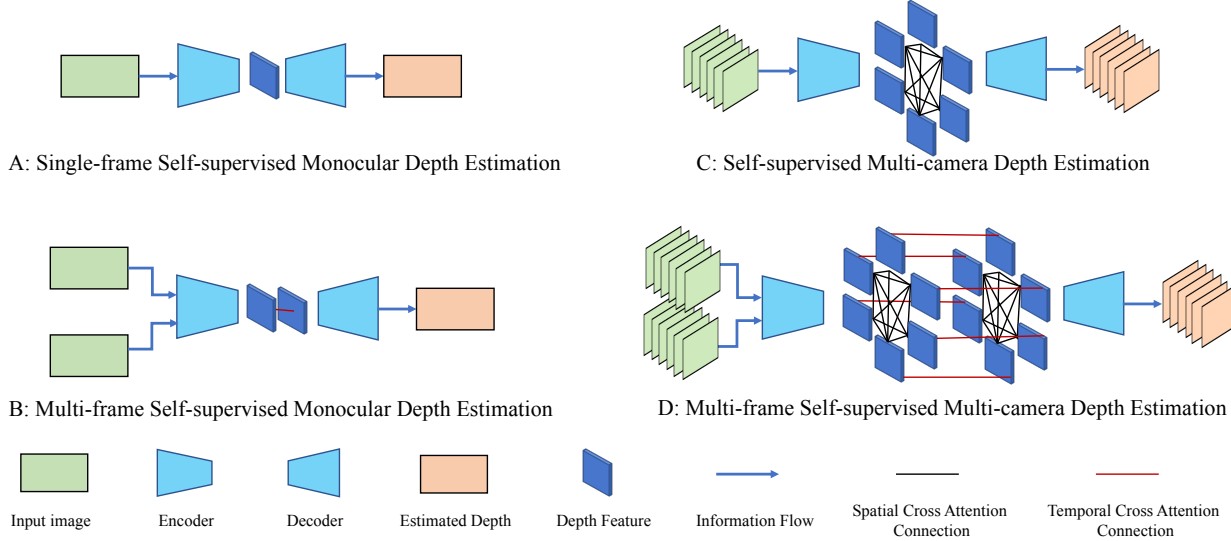

Figure 1: Comparison of self-supervised depth estimation pipelines under different settings (A: Single-frame Self-supervised Monocular Depth Estimation; B: Multi-frame Self-supervised Monocular Depth Estimation; C: Self-supervised Multi-camera Depth Estimation; D: Multi-frame Self-supervised Multi-camera Depth Estimation). Only depth networks are illustrated and the corresponding pose networks are omitted for simplicity.

are usually exploited to predict the corresponding depth and pose transformations, which are utilized to warp the source frame to adjacent frames and thereby optimize the networks by minimizing the photometric difference between the original images with the warped images. These approaches (Zhou et al., 2017; Godard et al., 2019; Zhao et al., 2022) employ multiple frames data solely during the training phase for the loss computation, and merely a single monocular image is taken as input during inference, thus can be categorized as Single-frame Self-supervised Monocular Depth Estimation methods as shown in Figure 1 A.

To leverage the readily available sequential image data effectively, some methods (Watson et al., 2021; Guizilini et al., 2022a; Zhang & Zhao, 2023) propose utilizing multi-frame images as input during both training and inference stages, shown as Figure 1 B. The inter-frame geometric correlations are usually exploited by constructing cost volumes or correlation layers. These approaches effectively enhance the performance of self-supervised depth estimation methods by harnessing the temporal multi-frame correlations.

In addition to the continually advancing self-supervised monocular depth estimation techniques, some approaches (Guizilini et al., 2022b; Wei et al., 2023; Xu et al., 2022a) are now extending monocular methods to the realm of multi-camera configurations to fulfill the perceptual requirements of autonomous driving cars encompassing 360-degree surround-view cameras. Based on the monocular counterpart, these methods additionally allow cross-camera feature interaction and fusion to utilize the overlap among adjacent cameras and boost the representation learning, shown as Figure 1 C. Taking the multi-camera sequence as input, the overlap of field-of-view (FoV) not only exists in adjacent cameras but also in adjacent temporal frames, which can be comprehensively exploited to facilitate depth representation learning, shown as Figure 1 D.

While adapting monocular self-supervised methods to the multi-camera setup has demonstrated promise in previous methods, some unique challenges specific to multi-camera setups are neglected, impeding further performance improvement. Self-supervised depth estimation methods highly rely on the co-visible regions among different frames to compute reprojection errors. Additionally, they assume that the projection pixels of the same 3D point in different images have identical intensities, which suffer from many violation cases e.g., illumination variance, extreme weather, occlusions, etc. For the multi-camera setting in large-scale autonomous driving datasets collected in the wild, e.g., NuScenes and DDAD, the challenges lie in 1) The overlap between adjacent cameras (like front camera w.r.t. front-left and front-right cameras) is too small (as low as 10% (Xu et al., 2022a)) to conduct effective image or feature matching for accurate 3D geometry

recovery; 2) There are various challenging weather or illumination cases including driving scenarios in rainy days or at night, affecting providing precise photometric supervision for self-supervised depth estimation.

In this paper, we dig into the Multi-frame Self-supervised Multi-camera Depth Estimation paradigm and propose novel techniques to mitigate these challenges and improve performance. We first propose a Spatial-Temporal Transformer to comprehensively exploit both local connectivity and the global context of image features, meanwhile learning enriched spatial-temporal cross-view correlations to recover 3D geometry. As shown in Figure 2, we exploit not only cross-camera correlation in the same frame (denoted by yellow arrows) and cross-frame correlation of the same camera (displayed as views with the corresponding same color) but also cross-camera and cross-frame correlation (different cameras in different frames, shown as colorful arrows among different temporal views) simultaneously. This strategy maximally utilizes the co-visibility overlap among images, thereby promoting both feature matching and network training. However, the illumination condition and brightness among cameras and frames could be variant as car driving, which is harmful to both image correlation acquisition and the projection error calculation in the self-supervised learning process. We thus leverage a spatial-temporal photometric consistency correction strategy to adjust the image intensities and maintain brightness consistency. Besides, we introduce a Generative Adversarial Network-based geometry regularization module to regularize the prediction weirdness in challenging cases, e.g. Rainy and Night scenarios.

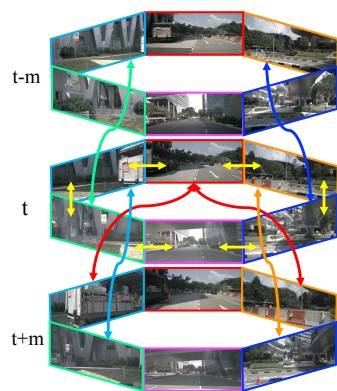

Figure 2: Illustration of simultaneous cross-camera and cross-frame correlations.

In summary, the main contributions of this paper are three-fold:

- We focus on the challenging self-supervised multi-camera depth estimation task by developing a novel Transformer-based framework.

- We propose a Spatial-Temporal Transformer (STTrans) for comprehensive feature extraction with further exploration of both cross-camera and cross-frame geometric correlations. Together with the spatial-temporal photometric consistency correction strategy (STPCC), our method can comprehensively utilize spatial-temporal context to enhance the depth and pose learning.

- We introduce an Adversarial Geometry Regularization (AGR) module to provide spatial positional restrictions for predicted depth maps, mitigating prediction weirdness in challenging cases, e.g. Rainy and Night scenarios.

- We conduct comprehensive evaluations and ablation studies, demonstrating the effectiveness of our method. It achieves state-of-the-art results on two large-scale self-supervised multi-camera depth estimation benchmarks, *i.e.* NuScenes (Caesar et al., 2020) and DDAD (Guizilini et al., 2020).

## 2 Related Work

### 2.1 Monocular Single-Frame Self-supervised Depth Estimation

Research into self-supervised depth estimation initially began with monocular settings, wherein researchers employed monocular image sequences as training data and estimated depth maps for individual monocular frames during inference. SfMLearner(Zhou et al., 2017) is one of the first attempts to explore monocular depth estimation in a self-supervised manner. It exploits predicted depth and pose to warp source images to reconstruct its adjacent images thereby formulating the learning as a projection error minimization process. Many subsequent works further improve this paradigm by additionally introducing 3D constraint (Mahjourian et al., 2018), imposing feature-level consistency (Shu et al., 2020), integrating uncertainty learning (Poggi et al., 2020; Yang et al., 2020) and incorporating related tasks, e.g., optical flow estimation (Yin

& Shi, 2018; Zhou et al., 2021; Zhao et al., 2020b) and semantic segmentation (Klingner et al., 2020; Jung et al., 2021). Monodepth2 (Godard et al., 2019) proposes several schemes to improve the effectiveness of photometric loss, including a minimum reprojection loss and an auto-masking strategy, yielding more accurate results. Recently, many works ( DIFFNet (Zhou et al., 2021), MonoFormer (Bae et al., 2022), MonoViT (Zhao et al., 2022) and SRD (Liu et al., 2023)) explore stronger network architectures to enhance the representation learning ability including PackNet (Guizilini et al., 2020), HRNet (Sun et al., 2019) and Vision Transformer (Dosovitskiy et al., 2020), further improves the prediction accuracy. **Besides, there is a line of work devoted to addressing illumination issues in adverse conditions such as nighttime driving scenarios (Vankadari et al., 2020; Wang et al., 2022; Zheng et al., 2023). Some methods (Vankadari et al., 2020; Wang et al., 2022) utilize domain adaptation techniques to adapt the daytime training estimation network to be applicable for the nighttime scenes. STEPS (Zheng et al., 2023) proposes to jointly learn a nighttime image enhancer and a depth estimator to overcome the low illumination problems in the depth estimation task, but additional illumination estimation and calibration networks are imposed, increasing computation burdens.**

## 2.2 Monocular Multi-Frame Self-supervised Depth Estimation

Given the availability of image sequences as training data, researchers then embarked on investigating how to leverage temporal information to further enhance the efficacy of monocular depth estimation. TC-Depth (Ruhkamp et al., 2021) fused the multi-frame features with proposed spatial and temporal attention modules to create a multi-frame depth estimation network, which improves the temporal depth stability and accuracy by combining modules with photometric cycle consistency. Inspired by multi-frame stereo methods (Kendall et al., 2017; Sun et al., 2018), ManyDepth (Watson et al., 2021) is introduced as an innovative self-supervised multi-frame depth estimation model that capitalizes on the synergies between monocular and multi-view depth estimation, incorporating multiple frames during the testing phase. DepthFormer (Guizilini et al., 2022a) proposed a novel end-to-end transformer, which generates cost volume through multi-view feature matching via cross- and self-attention with depth-discretized epipolar sampling. IterDepth (Feng et al., 2023) further improves the multi-frame monocular depth estimation approach with the proposed iterative residual refinement network, incorporating a gated recurrent depth fusion unit to enable iterative feature fusion and inverse depth prediction. DS-Depth (Miao et al., 2023) presents a dynamic cost volume leveraging residual optical flow to improve occlusion handling, further enhanced by a fusion module. Additionally, pyramid distillation and adaptive photometric error losses are proposed for accuracy improvement.

## 2.3 Multi-camera Self-supervised Depth Estimation

Multi-camera depth estimation is a long-standing topic, which is usually solved by multi-view stereo, *i.e.*, reconstructing 3D information of the scene from pictures of different angles. Multi-view stereo usually needs a large overlap to conduct image matching and cost volume construction, which is not suitable for driving scenes. FSM (Guizilini et al., 2022b) extends self-supervised monocular depth estimation to the surrounding multi-camera setting to meet the increasing demand in autonomous driving scenarios. FSM focuses on enhancing the self-supervision signal by leveraging spatial and temporal contexts to enrich the photometric consistency supervision and imposing pose consistency constraints to learn robust pose estimation. SurroundDepth (Wei et al., 2023) utilizes a shared encoder to extract high-level feature maps for each view with a cross-view transformer to fuse features and capture cross-view interactions. MCDP (Xu et al., 2022a) formulate the depth estimation as a weighted combination of depth basis to iteratively update and propagate to maintain a consistent structure of depth predictions. EGA-Depth (Shi et al., 2023) simplifies the cross-attention mechanism employed in SurroundDepth by limiting cross-attention to adjacent cameras for each individual camera. This refinement enables cross-attention to be conducted on higher-resolution features, further improving the accuracy.

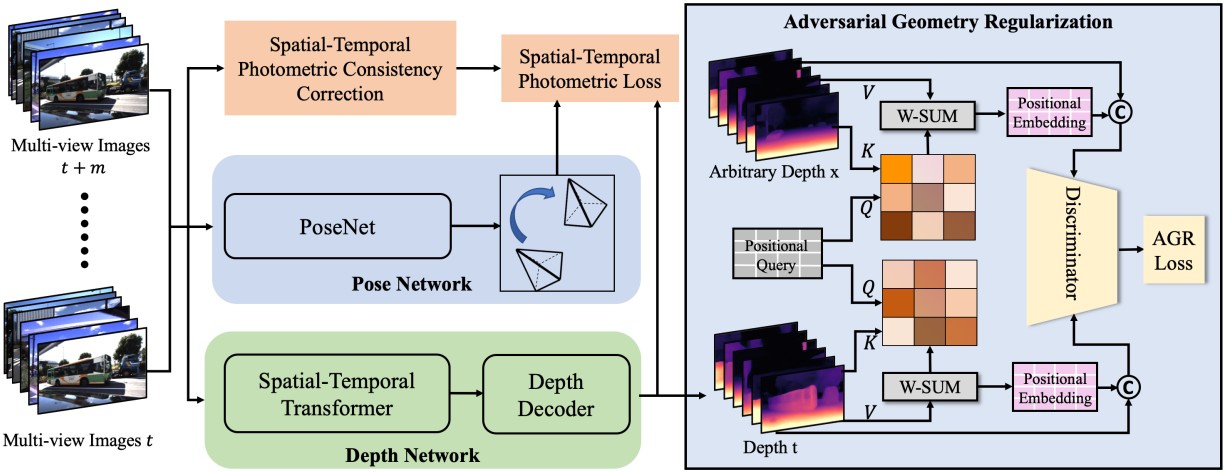

Figure 3: Overview of our STViT framework. Our STViT is composed of a Depth Network, a Pose Network, and an Adversarial Geometry Regularization Module. The Depth Network consists of a Spatial-Temporal Transformer Encoder and a Depth Decoder. The Pose Network is implemented by a lightweight ResNet. The Depth Network and Pose Network are jointly optimized via the minimization of Spatial-Temporal Photometric Loss. After predicted depth maps are obtained, they are further regularized and refined in the Adversarial Geometry Regularization Module.

## 2.4 Self-supervised Depth Estimation with Generative Adversarial Network

Generative Adversarial Network (GAN) (Goodfellow et al., 2014) has drawn broad attention in many vision tasks including style transfer (Jing et al., 2019; Xu et al., 2021), image-to-image translation (Isola et al., 2017; Zhu et al., 2017), image editing (Zhu et al., 2016; Chen et al., 2020; 2022), cross-domain image generation (Bousmalis et al., 2017; Deng et al., 2018), etc. Since our proposed Geometry Regularization Module is based on a Generative Adversarial Network (GAN), we review the previous self-supervised depth estimation with GAN. One line of work (CS Kumar et al., 2018; Zhao et al., 2020a; Xu et al., 2022b) utilizes GAN-based as a robust loss item to distinguish warped images and original images in the self-supervised depth estimation pipeline. Some other approaches (Zheng et al., 2018; Zhao et al., 2019; Sun et al., 2023) take the image-to-image translation ability of GAN to either enhance input image quality or transfer synthetic and realistic images to leverage additional synthetic datasets (Gaidon et al., 2016) to conduct domain adaptation. Wu et al. (Wu et al., 2019) and Wang et al. (Wang et al., 2021) also design a GAN-based module as a regularization and refinement. However, the former is dedicated to distinguishing the ground-truth depth map and the predicted depth map while the latter is devised to constrain the incorrect depth in nighttime with daytime prediction in an adversarial manner. Our work differs from these works in two aspects 1) We utilize arbitrary depth maps from other scenes to regularize the depth maps of the corresponding camera without corresponding ground truth or predictions in specific illumination conditions and 2) We design a novel depth-aware positional embedding together with predicted depth maps as the input of the discriminator instead of the corresponding RGB frames or coordinates.

## 3  Method

### 3.1  Network Architecture of STViT

#### 3.1.1  Motivation

In self-supervised depth estimation algorithms, there exists no explicit ground truth information, and the only available supervisory signal relies on photometric consistency across different viewpoints. In the case of a multi-camera setup with six cameras capturing temporal sequences, a wealth of data is available for

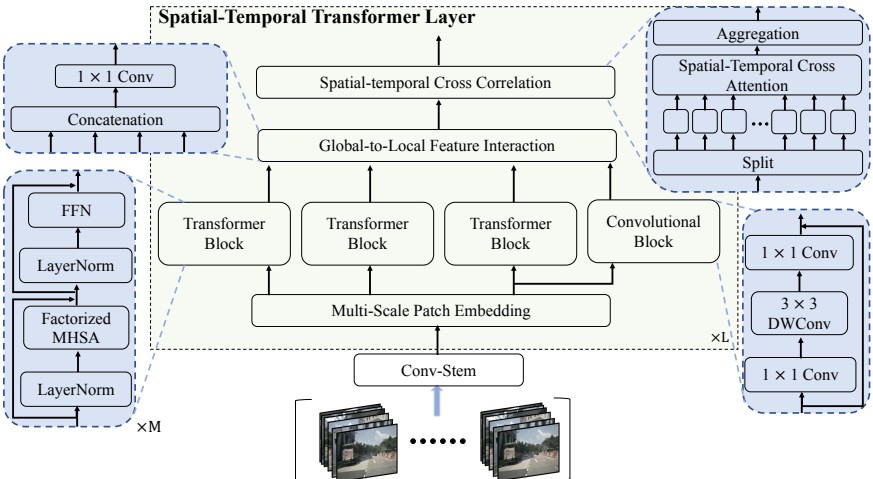

Figure 4: The architecture of Depth Encoder. It consists of Conv-Stem and Spatial-Temporal Transformer Layers. Each Transformer layer contains Multi-Scale Patch Embedding, Transformer Blocks, a Convolutional Block, a Global-to-Local Feature Interaction, and a Spatial-Temporal Cross Correlation Module. The structure of each module is illustrated in blue blocks.

training and inference. Consequently, depth estimation networks must proficiently extract both local and global features from the input images. This entails not only comprehensive feature extraction from individual frames but also the acquisition of geometric features across temporally sequential frames and co-observable regions among different camera viewpoints. Previous multi-camera self-supervised depth estimation methods typically employed Convolutional Neural Networks (CNNs) to extract features from input images and subsequently performed cross-attention operations explicitly between these image features. However, due to the localized nature of convolution operations, CNNs often struggle to capture long-range context similarity and dependencies effectively. Due to the excessively localized nature of the extracted features, which tend to focus on individual objects or semantic categories, even attempts to capture inter-frame correlations through subsequent cross-attention mechanisms have proven ineffective. This has hindered the accurate recovery of the geometric information of the entire scene (Zhao et al., 2022). Therefore, in order to better extract both global and local geometric features and leverage correlations across different viewpoints and sequential frames, we introduce the Spatial-Temporal Transformer Framework, referred to as STViT, which is specifically designed for multi-camera self-supervised depth estimation. The Framework follows the typical self-supervised depth estimation structure, consisting of a Depth Network and a Pose Network. The Depth Network is composed of a Spatial-Temporal Transformer and a Decoder.

### 3.1.2 Depth Network

Similar to prior works, our Depth Network is designed following the encoder-decoder architecture. We will explain the details of Depth Network in the following sections.

**Spatial-Temporal Transformer (STTrans)** Previous studies (Guizilini et al., 2020; Zhao et al., 2022) have highlighted the importance of extracting effective features to improve the performance of depth estimation. Therefore, we enhance the encoder architecture for multi-camera self-supervised depth estimation by employing powerful vision transformer models. We propose a Spatial-Temporal Transformer to not only leverage the transformer's ability to model long-range dependencies, overcoming the locality issue in feature extraction seen in previous works (Godard et al., 2019; Wei et al., 2023), but also introduce Spatial-Temporal Cross-Correlation to fully exploit the co-visibility regions across cameras and temporal frames for geometric structure recovery. Inspired by recent transformer models such as MPViT (Lee et al., 2022) which introduces the concept of a Multi-Path Transformer Block, we devise a Depth Encoder to capture both local and global context within images and further exploit the spatial-temporal cross correlations.

As shown in Figure 4, our Depth Encoder consists of Conv-Stem and Spatial-Temporal Transformer Layers. Each Transformer layer contains Multi-Scale Patch Embedding, Transformer Blocks, a Convolutional Block, a Global-to-Local Feature Interaction, and a Spatial-Temporal Cross Correlation Module. The input multi-camera sequence is fed to a Conv-Stem and then Spatial-Temporal Transformer Layers to obtain the depth feature. **The Spatial-Temporal Transformer layer first embeds the extracted features into different-sized visual tokens in Multi-Scale Patch Embedding which is formed by several parallel convolutional patch embedding layers with different kernel sizes, to exploit both fine- and coarse-grained visual tokens at the same feature level following MPViT (Lee et al., 2022)**. After that, parallel Transformer Blocks and Convolutional Block are leveraged to further process the embedded tokens. As shown in Figure 4, there are three Transformer Blocks to capture the long-range dependencies and global context. Each Block contains $M$ Transformer Layers, which consists of a Layer Normalization (LayerNorm) module, a Factorized Multi-head Self Attention (MHSA) layer (Lee et al., 2022), another Layer Normalization, and a Feed-forward Network (FFN). Parallel to the Transformer Blocks, a Convolutional Block is used to exploit local connectivity from translation invariance. The Convolutional Block comprises a sequence of $1 \times 1$, $3 \times 3$ depth-wise, and $1 \times 1$ convolutions. By combining the advantages of Transformer Blocks and Convolutional Blocks, the modeled feature can capture both local connectivity and global context simultaneously. A subsequent Global-to-Local Feature Interaction is further used to enhance the local and global feature interactions to obtain enriched representations. We use $Enc_{depth}$ to represent the feature extraction part of the model:

$$F = Enc_{depth}([I_t, ..., I_{t+m}]), \tag{1}$$

**where $I_t$ represents the surrounding six images at timestamp $t$ and $m \in \{-1, 0, 1\}$ means we take three temporal frames as input in our default setting.** Although we can effectively acquire the image feature, the cross-view correlation among different cameras and different temporal frames is still not exploited. Thus, we introduce a Spatial-Temporal Cross Correlation module to facilitate correlation learning and geometry recovery.

**Spatial-Temporal Cross Correlation** As shown in Figure 4, the interacted features are first split into different cameras and different temporal frames, e.g. $F_t^i$ denoting the feature of the $i$th camera in timestamps $t$. For each feature $F_t^i$, we pre-define the list of views that share overlap regions with feature $F_t^i$. The overlapped views contain adjacent cameras at the same timestamp, adjacent temporal frames of the same camera, and simultaneously cross-camera and cross-frame views as well, as shown in Figure 2. Thus, Spatial-Temporal Cross Correlation is able to learn enriched spatial-temporal cross-view correlations. Specifically, feature $F_t^i$ is leveraged to compute queries, and the features of overlapped views $G_{t+m}^x$ are used to obtain keys and values:

$$Q_t^i = F_t^i W_{q_t}^i, \ K_t^i = G_{t+m}^x W_{k_t}^i, \ V_t^i = G_{t+m}^x W_{v_t}^i. \tag{2}$$

Here, $x \in [1, N]$ and $N$ is the number of cameras. $W_{q_t}^i, W_{k_t}^i$, and $W_{v_t}^i$ are the learnable projections for query, key, and value. Thus, the feature after cross correlation is:

$$\hat{F}_t^i = softmax(\frac{Q_t^i K_t^i}{\sqrt{d}})V_t^i, \tag{3}$$

where $d$ denotes the embedding dimension. By interacting with features of both cross-camera and cross-frame views, features $\hat{F}_t^i$ can learn enriched cross-view context and correspondence, which is beneficial to accurately inferring 3D geometry.

**Depth Decoder** Utilizing multi-scale features obtained from the depth encoder, our depth decoder incorporates cross-layer and cross-scale connections following Zhao et al. (2022). Recognizing the contextual distinctions among features at various scales, such as the preference for higher-resolution features for fine-grained details, we bolster cross-scale feature integration. To achieve this, we employ both spatial and channel attention mechanisms. Ultimately, the disparity (inverse depth) predictions $D$ under different resolutions are handled by four heads comprising two convolutional layers and a Sigmoid activation function. The prediction of depth maps can be formulated as:

$$D_t^i = Dec_{depth}(\hat{F}_t^i), \tag{4}$$

where $Dec_{depth}$ is the decoder of depth network, $i$ denotes the $i$th camera and $t$ denotes the $t$th frame.

### 3.1.3 Pose Network

Following the common practice of self-supervised depth estimation, we adopt a lightweight ResNet18 (He et al., 2016) as the encoder $Enc_{pose}$ of Pose Network and subsequent convolution layers as the decoder $Dec_{pose}$ to regress the 6 DoF relative poses $P$ between adjacent temporal frames. Specifically, we first take $N$ pairs of adjacent frames as the input and output a universal pose $P_{t+m \to t}$ for all $N$ cameras (Wei et al., 2023). Then, the predicted universal pose $P_{s \to t}$ is transformed to each specific camera with its known camera extrinsic matrix. The whole process of predicting pose can be formulated as:

$$h^i = Enc_{pose}([I_t^i, I_{t+m}^i]),$$
$$P_{t+m \to t} = Dec_{pose}(\frac{1}{N} \sum_{i=1}^{N} h^i), \tag{5}$$
$$P_{t+m \to t}^i = (T^i)^{-1} P_{t+m \to t} T^i,$$

where $P_{t+m \to t}^i$ is the learned pose for the $i$th camera and $T^i$ is its corresponding extrinsic matrix. The universal pose prediction manner can naturally ensure geometry consistency among cameras.

### 3.2 Self-supervised Training

The self-supervised depth estimation problem is formulated to a projection error minimization process, where the depth network and pose network are jointly optimized. Given the input images, depth maps $D$ and relative pose transformation $P$ are predicted with depth network and pose network. Then the depth map and pose are utilized to reproject the source image to reconstruct the target image. The networks are optimized by minimizing the difference between the synthesized target image and the original target image.

### 3.2.1 Spatial-Temporal Photometric Loss

The target image can be each frame e.g., $I_t^i$ denoting the image captured by the $i$th camera at the timestamp $t$. To fully exploit the spatial-temporal consistency, the source images include not only the spatial neighborhood, the temporal neighborhood and also the cross-frame and cross-camera views with overlapped regions. Similar to the Spatial-Temporal Cross Correlation part, we pre-define a list of views that can be observed co-visible regions with the target image, e.g., the length of the $i$th correlation image list is $CI$. Thus, the photometric loss $\ell_p$ can be formulated as:

$$\ell_p = \sum_{i=1}^{N} \sum_{ci=1}^{CI} \ell_{ph}(I_t^i, I_{s \to t}^{ci}). \tag{6}$$

The reconstructed target image $I_{s \to t}^{ci}$ is obtained via reprojection with the predicted depth map $D_t^i$ and pose $P_{t \to s}^{ci}$:

$$I_{s \to t}^{ci} = Proj(K, P_{t \to s}^{ci}, D_t^i, K^{-1}, I_t^i), \tag{7}$$

where $K$ is the camera intrinsic matrix. The typical photometric loss in prior works comprises an SSIM (Wang et al., 2004) metric and L1 Loss term:

$$\ell_{ph}(I_t^i, I_{s \to t}^{ci}) = \alpha \frac{1 - SSIM(I_t^i, I_{s \to t}^{ci})}{2} + (1 - \alpha)||I_t^i - I_{s \to t}^{ci}||. \tag{8}$$

Moreover, an edge-aware smoothing term is often incorporated to add a regularization on depth maps in many previous works (Godard et al., 2017; 2019):

$$\ell_{sm} = |\partial_x \mu_{D_t}| e^{-|\partial_x I_t|} + |\partial_y \mu_{D_t}| e^{-|\partial_y I_t|}, \tag{9}$$

where $\mu_{D_t}$ is the inverse depth normalized by mean depth. $\partial_x \mu_{D_t}$ and $\partial_x \mu_{D_t}$ denote the disparity gradient among two directions.

### 3.2.2 Spatial-Temporal Photometric Consistency Correction (STPCC)

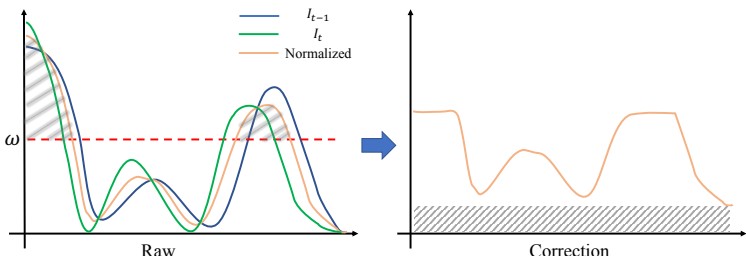

Figure 5: Illustration of the step of histogram adjusting in Spatial-Temporal Photometric Consistency Correction.

The photometric loss is designed based on the assumption that the same 3D points have the same intensity in diverse projected views. However, in practical outdoor driving scenarios, the illumination among different cameras and different timestamps can vary severely, which impedes network learning. Therefore, we propose Spatial-Temporal Photometric Consistency Correction (STPCC) to enforce the brightness consistency of diverse views before the calculation of photometric loss.

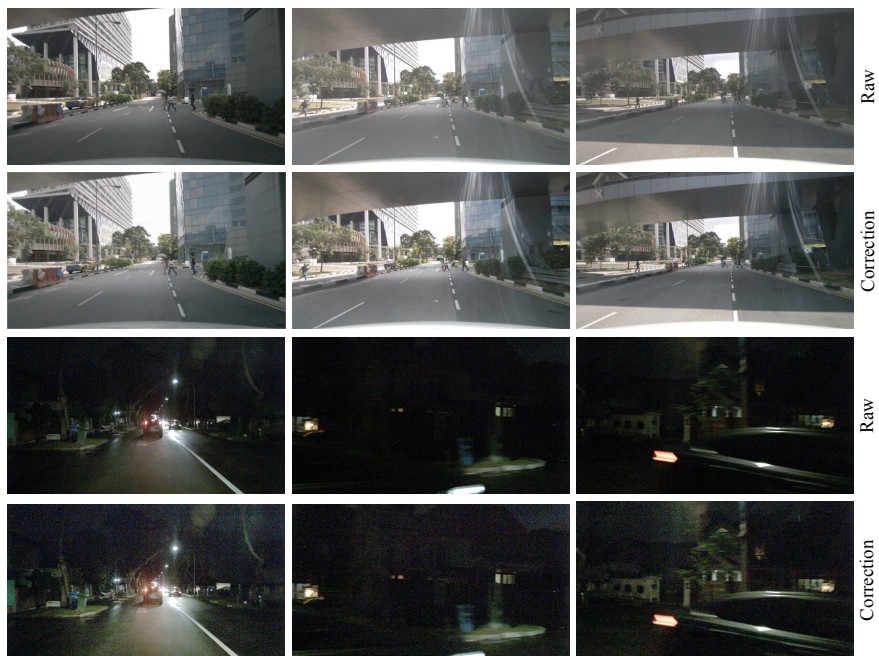

Figure 6: Qualitative display of images before and after Spatial-temporal Photometric Consistency Correction. The top two rows show the correction effect of three temporally adjacent images. The bottom two rows show the correction effect of spatially adjacent images in nighttime scenarios. Best viewed with zoom-in.

Inspired by Contrast Limited Histogram Equalication (CLHE) (Pizer et al., 1987), we leverage a common mapping function $\psi$ to correct image brightness and make the image color spatially and temporally consistent. We first compute the histograms $H$ of input images, which are the frequency distributions of $L$ intensity levels (usually $L \in \{0, 1, ..., 255\}$) of images. The histograms of spatially and temporally adjacent images, (taking temporal images as examples, $H_{t-1}, H_t, H_{t+1}$), are then processed by a normalization operation, $H = avg(H_{t-1}, H_t, H_{t+1})$. Based on the normalized frequency distribution, by setting a threshold $\omega$, we assume that if a certain intensity level in the histogram exceeds the threshold, it will be clipped, and the portions exceeding the threshold will be evenly distributed among the various intensity levels, as shown in Figure 5. After adjusting the Histograms consistently, the mapped figure (taking $I'_t$ as an example) can be obtained:

$$\bar{I}_t = \psi(H(I_t)) = \frac{CDF(H(I_t)) - CDF_{min}}{CDF_{max} - CDF_{min}} \times (|L|), \tag{10}$$

where $CDF$ represents the Cumulative Distribution Function. $CDF_{max}$ and $CDF_{min}$ are the corresponding maximum and minimum values of $CDF$.

In this way, the intensities distribution of spatially and temporally adjacent images can be aligned consistently. Moreover, the brightness of images in adverse illustration conditions *i.e.*, night or dark driving scenarios can be adjusted with higher visibility. The color correction effect is illustrated in Figure 6. The top two rows show the correction effect of three temporally adjacent images, which makes the brightness more consistent. The bottom two rows show the correction effect of spatially adjacent images in nighttime scenarios, which adjusts and improves the visibility. Note that STPCC is only applied to images before the photometric loss computation rather than the input for network learning. Therefore, the final photometric loss is:

$$\ell_p = \sum_{i=1}^{N} \sum_{ci=1}^{CI} \ell_{ph}(\bar{I}_t^i, \bar{I}_{s \to t}^{ci}). \tag{11}$$

### 3.3 Adversarial Geometry Regularization Module (AGR)

In real-world outdoor driving scenarios, adverse conditions such as rainy weather and nighttime driving are frequently encountered. Under such extreme circumstances, the effectiveness of photometric loss diminishes, thereby significantly affecting the performance of depth estimation. Therefore, we propose a GAN-based Adversarial Geometry Regularization Module (AGR) to further constrain the depth estimation, as shown in the right part of Figure 3. Specifically, we consider the Depth Network as a generator to provide depth map predictions. And adopt the depth predictions of an arbitrary normal-condition frame as a reference to regularize the depth distribution. It is observed the depth value distribution has a close relationship with the pixel positions (Dijk & Croon, 2019). Thus, we use the positional query to scan over the depth map which serves as key and value. So that we can obtain the depth-aware positional embedding $e_t^i$ by calculating the dot product similarity between the query and keys. In this way, the depth-aware positional embedding can provide soft geometric correspondence between query positions and depth maps. After that, the positional embedding is concatenated with the normalized predicted depth maps, denoted as $[e_t^i, \mu(D_t^i)]$. Similarly, the arbitrary depth maps are also concatenated with the corresponding positional embedding, denoted as $[e_t^{iR}, \mu(D_t^{iR})]$. We use the **PatchGAN (Isola et al., 2017) discriminator** $\Theta_{Dis}$ to distinguish $[e_t^i, \mu(D_t^i)]$ and $[e_t^{iR}, \mu(D_t^{iR})]$, while the depth network tries to make the prediction $[e_t^i, \mu(D_t^i)]$ indistinguishable with the regularization reference $[e_t^{iR}, \mu(D_t^{iR})]$. **The PatchGAN network consists of 5 layers, which progressively extract features from the input image. Each convolutional layer is followed by a LeakyReLU activation function, introducing non-linearity to the network. Batch normalization layers are inserted after every other convolutional layer to stabilize and speed up training. The final layer of the network is a separate convolutional layer and the output patch size is $1/8$ times the original depth predictions. Overall, the network gradually reduces the spatial dimensions of the input while increasing the number of feature channels, culminating in a classification output with two units corresponding to the desired classes.**

The optimization objective for AGR can be formulated as:

$$
\begin{aligned}
L_{Dis} =& \frac{1}{2}\mathbb{E}_{D_t^{iR}}[(\Theta_{Dis}([e_t^{iR}, \mu(D_t^{iR})]) - 1)^2] + \frac{1}{2}\mathbb{E}_{D_t^i}[\Theta_{Dis}([e_t^i, \mu(D_t^i)])^2], \\
L_{Gen} =& \frac{1}{2}\mathbb{E}_{D_t^i}[(\Theta_{Dis}([e_t^i, \mu(D_t^i)]) - 1)^2], \\
L_{AGR} =& \min_{Gen} \max_{Dis} L_{Dis} + L_{Gen}.
\end{aligned}
\tag{12}
$$

### 3.4 Training Loss

To sum up, the final training loss consists of the photometric loss $\ell_p$ (Eq. 11), the smoothing loss $\ell_{sm}$ (Eq. 9) and the AGR regularization loss $\ell_{AGR}$ (Eq. 12):

$$Loss = \ell_p + 10^{-3}\ell_{sm} + 5 \times 10^{-4}\ell_{AGR}. \tag{13}$$

Here, the loss weights of the photometric loss and the smoothing loss are kept the same as the monocular depth estimation methods while the parameter of the AGR regularization loss is obtained by empirical experiments.

## 4 Experiment

### 4.1 Datasets

Following the common practice in previous multi-camera depth estimation methods, we adopted NuScenes (Caesar et al., 2020) and DDAD (Guizilini et al., 2020) to evaluate our method. These two recently released autonomous driving datasets are both with six surrounded cameras and relatively small overlaps among cameras, which are more challenging than the prior monocular datasets.

**NuScenes** The NuScenes dataset (Caesar et al., 2020) encapsulates urban driving contexts and is characterized by a coordinated assemblage of imagery acquired from a sextuple-camera configuration. This compilation encompasses 1,000 distinct scenes and boasts an extensive repository of 1.4 million images. Renowned for its role as a benchmark for diverse tasks encompassing 2D and 3D object detection, alongside semantic and instance segmentation, this dataset assumes a pivotal position in the domain. Particularly pertinent to the self-supervised depth estimation task, the NuScenes dataset poses inherent challenges attributed to the relatively modest image resolution, constrained spatial inter-camera overlap, variegated weather conditions, diurnal temporal variations, and complex, unstructured settings. The raw image dimensions are specified as $1600 \times 900$, subsequently downscaled to a resolution of $640 \times 352$. Captured at a frequency of 30Hz, dataset samples are annotated at a reduced 2Hz cadence, dictated by keyframes. The temporal interval between these key frames is appreciably large, precluding the training of deep networks through conventional self-supervision techniques. Consequently, annotated Sweep data emerge as a viable recourse, furnishing pivotal supervisory signals in the training process.

**DDAD** The Dense Depth for Automated Driving (DDAD) dataset (Guizilini et al., 2020) encompasses urban driving scenarios and has been meticulously recorded through six synchronized cameras, displaying limited spatial overlap. It is distinguished by its provision of highly precise dense ground-truth depth maps for evaluative purposes, extending up to an impressive maximum depth range of 250 meters. This dataset comprises a training subset encompassing 12,650 instances (comprising 63,250 images) and a validation subset containing 3,950 instances (consisting of 15,800 images). In the training set, the utilization of ground-truth depth maps is eschewed. Notably, the image resolution is denoted as $1,936 \times 1,216$, following which, in consonance with the methodology delineated in [16], input images undergo a downsampling procedure to achieve a resolution of $640 \times 384$. Subsequently, during the evaluation phase, image resolution is restored to its original dimensions through bilinear interpolation.

### 4.2 Evaluation Metrics

The evaluation metrics for multi-camera depth estimation are the same as its monocular counterpart. Four error metrics: **Abs Rel** for Absolute Relative Error, **Sq Rel** for Square Relative Error, **RMSE** for Root Mean Square Error, **RMSE log** for Root Mean Square Logarithmic Error and three accuracy metrics are included:

- *Abs Rel* $= (1/n) \sum_{i \in n} ((|d_i - d_i^*|)/d_i)$,

- *Sq Rel* $= (1/n) \sum_{i \in n} ((||d_i - d_i^*||^2)/d_i)$,

- *RMSE* $= ((1/n) \sum_{i \in n} ||d_i - d_i^*||^2)^{1/2}$,

- *RMSE log* $= ((1/n) \sum_{i \in n} ||log(d_i) - log(d_i^*)||^2)^{1/2}$

- Accuracy: % of $d_i$ s.t. $max((d_i/d_i^*), (d_i^*/d_i)) = \delta < \delta_n$,

| Methods | Resolution | Abs Rel ↓ | Sq Rel ↓ | RMSE ↓ | RMSE log ↓ | $\delta < 1.25$ ↑ | $\delta < 1.25^2$ ↑ | $\delta < 1.25^3$ ↑ |
|---|---|---|---|---|---|---|---|---|
| Monodepth2 (Godard et al., 2019) | $352 \times 640$ | 0.287 | 3.349 | 7.184 | 0.345 | 0.641 | 0.845 | 0.925 |
| PackNet-SfM (Guizilini et al., 2020) | $352 \times 640$ | 0.309 | 2.891 | 7.994 | 0.345 | 0.547 | 0.796 | 0.899 |
| FSM* (Guizilini et al., 2022b) | $352 \times 640$ | 0.334 | 2.845 | 7.786 | 0.406 | 0.508 | 0.761 | 0.894 |
| SurroundDepth (Wei et al., 2023) | $352 \times 640$ | $0.245_{\pm 0.002}$ | $3.067_{\pm 0.006}$ | $6.835_{\pm 0.004}$ | $0.321_{\pm 0.001}$ | $0.719_{\pm 0.002}$ | $\underline{0.878}_{\pm 0.001}$ | $0.935_{\pm 0.001}$ |
| MCDP (Xu et al., 2022a) | $448 \times 768$ | 0.237 | 3.030 | 6.822 | - | 0.719 | - | - |
| EGA-Depth (Shi et al., 2023) | $352 \times 640$ | 0.239 | **2.357** | 6.801 | 0.317 | $\underline{0.723}$ | **0.880** | $\underline{0.936}$ |
| STViT (single) | $352 \times 640$ | $\underline{0.235}_{\pm 0.001}$ | $2.934_{\pm 0.005}$ | $\underline{6.736}_{\pm 0.003}$ | $\underline{0.315}_{\pm 0.001}$ | $\mathbf{0.724}_{\pm 0.001}$ | $0.877_{\pm 0.001}$ | $\underline{0.936}_{\pm 0.001}$ |
| STViT | $352 \times 640$ | $\mathbf{0.233}_{\pm 0.001}$ | $\underline{2.815}_{\pm 0.004}$ | $\mathbf{6.681}_{\pm 0.003}$ | $\mathbf{0.312}_{\pm 0.001}$ | $\mathbf{0.724}_{\pm 0.001}$ | $\underline{0.878}_{\pm 0.001}$ | $\mathbf{0.937}_{\pm 0.001}$ |

Table 1: Quantitative evaluation of self-supervised multi-camera depth estimation on nuScenes (Caesar et al., 2020). The best results are highlighted in bold. The row of FSM* shows the results of FSM reproduced by (Wei et al., 2023). The best results in each column are highlighted in **bold**, while the second-best ones are underlined. **The error bar is displayed in red color, summarized from 5 times inference.**

where $n$ is the total number of pixels in the ground truth depth map, $d_i$ and $d_i^*$ represent the predicted and ground truth depth value of pixel $i$. $\delta_n$ denotes a threshold, which is usually set to $1.25^1$, $1.25^2$ and $1.25^3$.

### 4.3 Implementation Details

We implement our STViT in Pytorch. The model is trained for 5 epochs on the NuScenes dataset (Caesar et al., 2020) and 20 epochs on the DDAD dataset (Guizilini et al., 2020) using AdamW as the optimizer and a batch size set to 6. The initial learning rate for PoseNet and depth decoder is $10^{-4}$, while the Transformer-based depth encoder is trained with an initial learning rate of $5 \times 10^{-5}$. Both the pose encoder and depth encoder are pre-trained on ImageNet (Dosovitskiy et al., 2020). We use 4 A100 GPUs for the experiments on Nuscenes and 8 GPUs for experiments on DDAD. In our experiments, we adopt the same data augmentation detailed in (Godard et al., 2019; Zhao et al., 2022). For our default setting, we use 3 temporal frames as input and we also test the version with a single temporal input.

### 4.4 Comparison with the state-of-the-arts

We conduct extensive quantitative evaluations on two large-scale autonomous driving datasets, *i.e.*, Nuscenes (Caesar et al., 2020) and DDAD (Guizilini et al., 2020) datasets. Our method is compared with two approaches adapted from monocular depth estimation methods (Godard et al., 2019; Guizilini et al., 2020) and four state-of-the-art multi-camera-based methods (Guizilini et al., 2022b; Wei et al., 2023; Xu et al., 2022a; Shi et al., 2023). The detailed evaluation results are presented in Tables 1 and 2. In comparison with recent state-of-the-art methods (Wei et al., 2023; Xu et al., 2022a; Shi et al., 2023), our approach demonstrates superior performance across most evaluation metrics, achieving the best results in five out of seven metrics on Nuscenes and four out of seven on DDAD. Our method leverages multiple temporal sequences input in the Spatial-Temporal Transformer, and for completeness, we also showcase its performance with a single temporal input (six camera figures at the same timestamp). Despite a slight performance degradation without temporal input and modeling, our single-input version still delivers promising results compared to other advanced methods.

### 4.5 Ablation Study

#### 4.5.1 Performance of individual cameras

To provide a comprehensive understanding of inference performance, we present an extensive presentation of evaluation results concerning the six individual cameras in both the Nuscenes and DDAD datasets, detailed in Tables 3 and 4, respectively. The experiment reveals that self-supervised depth estimation performs exceptionally well on front views compared to back views. Furthermore, the inference results in the left view significantly outperform their right counterpart. This divergence might be attributed to the inherent dissimilarities in scenes captured on opposing sides, signifying the sensitivity of the model to the specific spatial characteristics within its field of vision. This detailed examination and analysis may shed light on the intricacies of its responses to diverse perspectives, contributing valuable insights for future refinement in model designation and learning strategies.

| Methods | Resolution | Abs Rel ↓ | Sq Rel ↓ | RMSE ↓ | RMSE log ↓ | $\delta < 1.25$ ↑ | $\delta < 1.25^2$ ↑ | $\delta < 1.25^3$ ↑ |
|---|---|---|---|---|---|---|---|---|
| Monodepth2 (Godard et al., 2019) | 384 × 640 | 0.217 | 3.641 | 12.962 | 0.323 | 0.699 | 0.877 | 0.939 |
| PackNet-SfM (Guizilini et al., 2020) | 384 × 640 | 0.234 | 3.802 | 13.253 | 0.331 | 0.672 | 0.860 | 0.931 |
| FSM* (Guizilini et al., 2022b) | 384 × 640 | 0.229 | 4.589 | 13.520 | 0.327 | 0.677 | 0.867 | 0.936 |
| SurroundDepth (Wei et al., 2023) | 384 × 640 | $0.200_{\pm0.002}$ | $3.392_{\pm0.004}$ | $12.270_{\pm0.004}$ | $0.301_{\pm0.002}$ | $0.740_{\pm0.001}$ | $0.894_{\pm0.001}$ | $0.947_{\pm0.001}$ |
| MCDP (Xu et al., 2022a) | 384 × 640 | 0.193 | 3.111 | 12.264 | - | **0.811** | - | - |
| EGA-Depth (Shi et al., 2023) | 384 × 640 | 0.195 | 3.211 | **12.117** | 0.297 | 0.743 | **0.896** | 0.947 |
| STViT (single) | 384 × 640 | $0.193_{\pm0.001}$ | $3.093_{\pm0.002}$ | $12.206_{\pm0.002}$ | $0.295_{\pm0.001}$ | $0.735_{\pm0.001}$ | $0.895_{\pm0.001}$ | $0.948_{\pm0.001}$ |
| STViT | 384 × 640 | $\mathbf{0.192}_{\pm0.001}$ | $\mathbf{2.965}_{\pm0.002}$ | $12.156_{\pm0.003}$ | $\mathbf{0.293}_{\pm0.001}$ | $0.734_{\pm0.001}$ | $0.895_{\pm0.001}$ | $\mathbf{0.949}_{\pm0.001}$ |

Table 2: Quantitative evaluation of self-supervised multi-camera depth estimation on DDAD (Guizilini et al., 2020). The best results are highlighted in bold. The row of FSM* shows the results of FSM reproduced by (Wei et al., 2023). The best results in each column are highlighted in **bold**, while the second-best ones are underlined. **The error bar is displayed in red color, summarized from 5 times inference.**

| Cameras | Resolution | Abs Rel ↓ | Sq Rel ↓ | RMSE ↓ | RMSE log ↓ | $\delta < 1.25$ ↑ | $\delta < 1.25^2$ ↑ | $\delta < 1.25^3$ ↑ |
|---|---|---|---|---|---|---|---|---|
| Front | 352 × 640 | 0.153 | 1.845 | 7.108 | 0.245 | 0.803 | 0.928 | 0.968 |
| Front-Left | 352 × 640 | 0.231 | 2.186 | 6.322 | 0.313 | 0.710 | 0.868 | 0.931 |
| Back-Left | 352 × 640 | 0.231 | 2.233 | 5.825 | 0.312 | 0.727 | 0.869 | 0.930 |
| Back | 352 × 640 | 0.193 | 2.277 | 7.286 | 0.292 | 0.741 | 0.901 | 0.954 |
| Back-Right | 352 × 640 | 0.304 | 4.372 | 6.569 | 0.358 | 0.676 | 0.846 | 0.918 |
| Front-Right | 352 × 640 | 0.286 | 3.980 | 6.974 | 0.352 | 0.688 | 0.858 | 0.922 |
| All | 352 × 640 | 0.233 | 2.815 | 6.681 | 0.312 | 0.724 | 0.878 | 0.937 |

Table 3: Quantitative evaluation of corresponding six cameras of self-supervised multi-camera depth estimation on Nuscenes (Caesar et al., 2020).

### 4.5.2 Ablation study for proposed contributions

To demonstrate the effectiveness of each component of our methods, we conduct thorough ablation studies on both the Nuscenes and DDAD datasets, with detailed findings presented in Table 5 and Table 6. Utilizing SurroundDepth as our baseline, we systematically introduce and evaluate each augmentation, including the Spatial-Temporal Transformer (STTrans), Spatial-Temporal Photometric Consistency Correction (STPCC), and the Adversarial Geometry Regularization module (AGR). The performance trends observed across the datasets exhibit consistent variations. The Spatial-Temporal Transformer notably enhances depth estimation outcomes, leveraging improved feature extraction and spatial-temporal cross-view feature interaction. STPCC, augmenting the photometric loss calculation through adjustments in the alignment of multiple spatial-temporal input images, brings further enhancements, as evidenced in Table 5 and Table 6. Moreover, the Adversarial Geometry Regularization module, denoted as AGR, significantly reduces prediction errors, validating its efficacy. To offer a more vivid illustration of the impact of AGR, we conduct a qualitative ablation by visualizing predicted depth maps both with and without the inclusion of AGR in the model, as depicted in Figure 7. The comparison showcases that the model without AGR tends to generate artifacts in challenging conditions such as low-illumination regions. In contrast, our complete model incorporating AGR effectively mitigates these issues, underscoring the crucial role of AGR in enhancing the robustness of the model, especially in adverse conditions.

| Cameras | Resolution | Abs Rel ↓ | Sq Rel ↓ | RMSE ↓ | RMSE log ↓ | $\delta < 1.25$ ↑ | $\delta < 1.25^2$ ↑ | $\delta < 1.25^3$ ↑ |
|---|---|---|---|---|---|---|---|---|
| Front | 384 × 640 | 0.130 | 2.699 | 13.219 | 0.216 | 0.845 | 0.945 | 0.977 |
| Front-Left | 384 × 640 | 0.186 | 2.745 | 11.845 | 0.294 | 0.745 | 0.898 | 0.948 |
| Back-Left | 384 × 640 | 0.199 | 2.885 | 11.419 | 0.301 | 0.729 | 0.891 | 0.944 |
| Back | 384 × 640 | 0.188 | 3.062 | 14.027 | 0.292 | 0.717 | 0.900 | 0.956 |
| Back-Right | 384 × 640 | 0.224 | 3.021 | 10.874 | 0.331 | 0.683 | 0.866 | 0.935 |
| Front-Right | 384 × 640 | 0.224 | 3.377 | 11.552 | 0.327 | 0.684 | 0.867 | 0.934 |
| All | 384 × 640 | 0.192 | 2.965 | 12.156 | 0.293 | 0.734 | 0.895 | 0.949 |

Table 4: Quantitative evaluation results of corresponding six cameras of self-supervised multi-camera depth estimation on DDAD (Guizilini et al., 2020).

| Methods | Abs Rel ↓ | Sq Rel ↓ | RMSE ↓ | RMSE log↓ |
|---|---|---|---|---|
| Baseline | 0.245 | 3.067 | 6.835 | 0.321 |
| + STTrans | 0.238 | 2.889 | 6.732 | 0.316 |
| + STTrans + STPCC | 0.236 | 2.864 | 6.709 | 0.315 |
| + STTrans + STPCC + AGR | 0.233 | 2.815 | 6.681 | 0.312 |

Table 5: Ablation study on Nuscenes (Caesar et al., 2020). STTrans denotes the Spatial-Temporal Transformer framework and AGR represents our Adversarial Geometry Regularization module.

| Methods | Abs Rel ↓ | Sq Rel ↓ | RMSE ↓ | RMSE log↓ |
|---|---|---|---|---|
| Baseline | 0.200 | 3.392 | 12.270 | 0.301 |
| + STTrans | 0.195 | 3.126 | 12.204 | 0.297 |
| + STTrans + STPCC | 0.194 | 3.103 | 12.189 | 0.295 |
| + STTrans + STPCC + AGR | 0.192 | 2.965 | 12.156 | 0.293 |

Table 6: Ablation study on DDAD (Guizilini et al., 2020). STTrans denotes the Spatial-Temporal Transformer framework and AGR represents our Adversarial Geometry Regularization module.

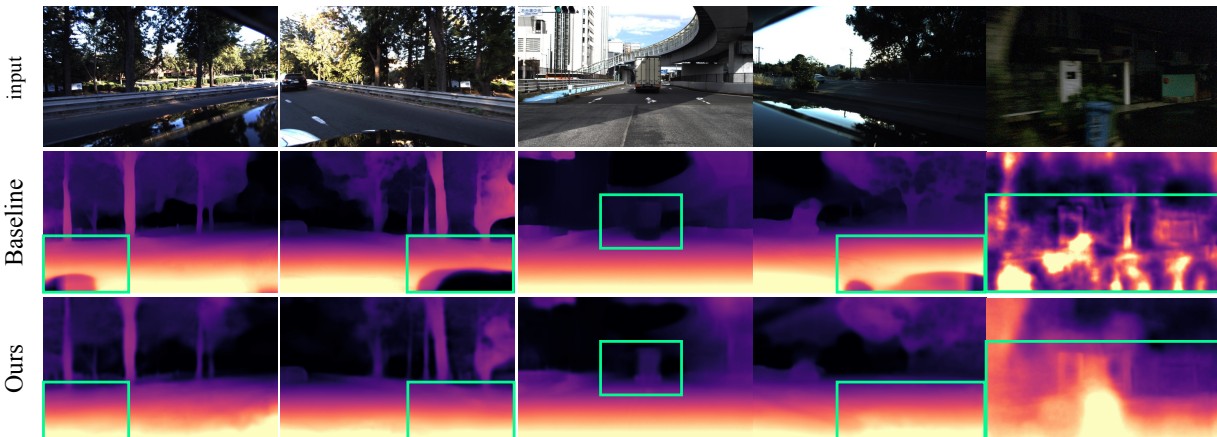

Figure 7: **Qualitative ablation of AGR. Regions with large differences are highlighted with green boxes. The visualization comparison can demonstrate the effectiveness of AGR in constraining prediction weirdness in low-illumination and nighttime driving scenarios.**

### 4.5.3 Ablation study for Spatial-Temporal Transformer (STTrans)

**Ablation study of structure.** The ablation study conducted on the Spatial-Temporal Transformer (STTrans) structure, detailed in Table 7 and Table 8, provides insights into the critical components influencing its performance. The variants explored include modifications to the structure components as illustrated in Figure 4, including Convolutional Block (CNN Path), Transformer Block (Trans. Path), adjustments in the number of Transformer Block Paths, and alterations in the structure of the Spatial-Temporal Cross-Correlation (STCC). Examining Table 7, it is evident that both the Convolutional Block and Transformer Block significantly contribute to the feature extraction process.

**Ablation study of spatial-temporal cross correlation.** In Table 8, specific analyses involve the removal of the full STCC, spatial cross-correlation (SCC), and temporal cross-correlation (TCC). The outcomes underscore the indispensability of both spatial and temporal cross-correlation mechanisms. Notably, the Convolutional Block and Transformer Block act as pivotal elements in shaping the feature representation, while the inclusion of spatial and temporal cross-correlation mechanisms enhances the model's capacity for capturing intricate spatial-temporal dependencies. These findings emphasize the separate effectiveness and interplay of components within the STTrans architecture, highlighting its holistic design for effective multi-camera depth estimation in driving scenarios.

| Methods | Abs Rel ↓ | Sq Rel ↓ | RMSE ↓ | RMSE log ↓ |
|---|---|---|---|---|
| CNN Path only | 0.256 | 3.418 | 8.675 | 0.301 |
| Trans. Path only | 0.248 | 3.272 | 8.016 | 0.398 |
| 1 Trans. Path | 0.246 | 3.165 | 7.693 | 0.346 |
| 2 Trans. Path | 0.243 | 3.134 | 7.238 | 0.325 |
| STTrans | 0.233 | 2.815 | 6.681 | 0.312 |

Table 7: Ablation study of Spatial-Temporal Transformer (STTrans) on Nuscenes (Caesar et al., 2020). CNN Path and Trans. Path denote the Convolutional Block and Transformer Block, respectively.

| Methods | Abs Rel ↓ | Sq Rel ↓ | RMSE ↓ | RMSE log ↓ |
|---|---|---|---|---|
| w/o STCC | 0.242 | 2.986 | 6.985 | 0.321 |
| w/o SCC | 0.238 | 2.956 | 6.893 | 0.318 |
| w/o TCC | 0.235 | 2.934 | 6.736 | 0.315 |
| STTrans | 0.233 | 2.815 | 6.681 | 0.312 |

Table 8: Ablation study of Spatial-Temporal Transformer (STTrans) on Nuscenes (Caesar et al., 2020). SCC and TCC denote the spatial cross-correlation and temporal cross-correlation, respectively.

**Ablation study of overlapping proportion.** Overlap regions are very critical in self-supervised depth estimation in two aspects, cross-view correlation and photometric loss calculation. To explore the significance of overlapping regions, we conduct an ablation experiment by applying a mask to exclude different proportions of the overlap area. As shown in Figure 4.5.3, we illustrate the overlap region with red lines and the applied mask with blue blocks, taking 1/3 masking in Front-Left, Front, and Front-right cameras as an example. According to the experiment results in Table 4.5.3, model performance degrades as the proportion of the overlap area decreases, verifying the value of view overlaps.

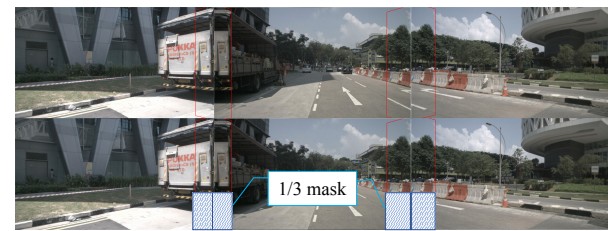

Figure 8: Illustration of spatial overlap regions (red lines) and the applied masks (blue block).

| Overlap | Abs Rel ↓ | Sq Rel ↓ | RMSE ↓ | RMSE log ↓ |
|---|---|---|---|---|
| 0 | 0.248 | 3.028 | 7.013 | 0.336 |
| 1/3 | 0.240 | 2.962 | 6.906 | 0.320 |
| 2/3 | 0.236 | 2.952 | 6.738 | 0.317 |
| 1 | 0.233 | 2.815 | 6.681 | 0.312 |

Table 9: Ablation study of remained overlapping proportions, including 0%, 1/3, 2/3 and 100%, after being excluded with masks.

### 4.5.4 Ablation study for Adversarial Geometry Regularization (AGR)

We extend our exploration to the position embedding approach within the Adversarial Geometry Regularization (AGR) module, conducting an insightful ablation study. In our analysis, we introduce a variant denoted as AGR (w/ concat), inspired by the methodology presented in the work by (Wang et al., 2021). This variant integrates arbitrary depth maps and 2D pixel coordinates through a concatenation process. The ablation results, outlined in Table 10, showcase the distinct performances of these approaches. Notably, our proposed depth-aware positional embedding operation demonstrates superior efficacy compared to the simpler concatenation strategy, affirming the significance of our design choice in enhancing the overall performance of the AGR module. This observation reinforces the critical role of thoughtful positional embedding strategies in optimizing depth estimation under adverse conditions within the self-supervised multi-camera context.

### 4.6 Qualitative Evaluation Results

In Figure 9, we present the qualitative evaluation results, showcasing the effectiveness of our proposed method. The top four rows depict input images and the corresponding predicted depth maps from the Nuscenes dataset, while the bottom four rows showcase analogous results from the DDAD dataset. The visual inspection of these results underscores the capability of our method to generate high-quality depth maps. Notably, our approach excels in capturing fine contextual details and delineating clear borders around objects. This qualitative assessment provides a compelling visual demonstration of the robustness and accuracy of our depth estimation method across diverse scenes and datasets.

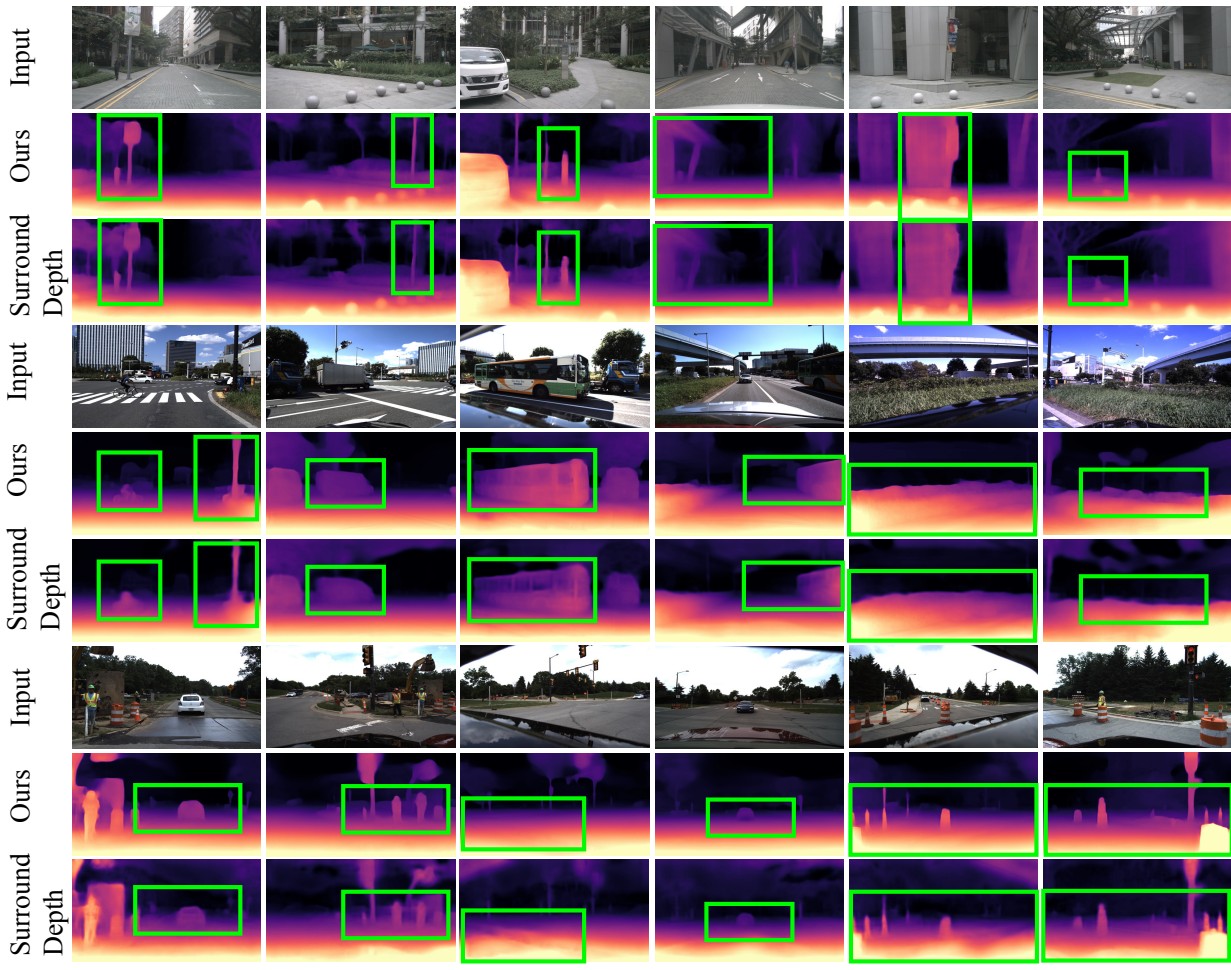

Figure 9: **Qualitative evaluation results and comparison with other state-of-the-art methods on Nuscenes (top three rows) and DDAD (bottom six rows). For each scene, we show the front, front-left, back-left, back, back-right, and front-right camera views from left to right. The predicted depth maps of our methods on both datasets display flatter ground, clearer object contour, and finer texture details, as highlighted in green boxes.**

| Methods | Abs Rel ↓ | Sq Rel ↓ | RMSE ↓ | RMSE log ↓ |
|---|---|---|---|---|
| AGR (w/ concat) | 0.235 | 2.851 | 6.697 | 0.314 |
| AGR | 0.233 | 2.815 | 6.681 | 0.312 |

Table 10: Ablation study of AGR on Nuscenes (Caesar et al., 2020). AGR (w/ concat) means directly using the concatenation of depth maps and positions.

| Methods | Abs Rel ↓ | RMSE ↓ | GFLOPs ↓ |
|---|---|---|---|
| **SurroudDepth** | 0.245 | 6.835 | 132.32 |
| **SurroudDepth-T** | 0.368 | 7.315 | 220.15 |
| **EGA-Depth** | 0.239 | 6.801 | 64.94 |
| **EGA-Depth-T** | 0.237 | 6.769 | 91.56 |
| **Ours(single)** | 0.235 | 6.736 | 68.66 |
| **Ours** | 0.233 | 6.681 | 96.80 |

Table 11: **Comparison of model computational efficiency.**

### 4.7 Model Computational Efficiency

**To investigate the impact of model computational requirements on model performance, we compare our method with other state-of-the-art methods in Table 11. "SurroudDepth-T" and**

**"EGA-Depth-T" are the corresponding variants of state-of-the-art methods utilizing multiple temporal frames. According to the results in Table 11, our method can achieve better performance (4% and 36% improvement on SurroundDepth and SurroundDepth-T; 1.6% and 1.7% improvement on EGA-Depth and EGA-Depth-T) without comparable computation requirements.**

## 5 Conclusion

This paper has addressed the intricate challenges of self-supervised multi-camera depth estimation in the context of autonomous driving, presenting a novel approach through our proposed Transformer-based framework, STViT. Our framework's Spatial-Temporal Transformer (STTrans) effectively harnesses local spatial connectivity and global context to recover rich spatial-temporal cross-view correlations, facilitating the intricate task of 3D geometry recovery. The incorporation of a spatial-temporal photometric consistency correction strategy (STPCC) further enhances performance by mitigating the impact of varying illumination conditions. Moreover, our Adversarial Geometry Regularization (AGR) module, rooted in Generative Adversarial Networks, introduces valuable spatial positional constraints, significantly improving depth map predictions in challenging scenarios, such as adverse weather and night driving conditions. The demonstrated superior performance of our method on large-scale datasets, including NuScenes and DDAD, reaffirms its efficacy and places it at the forefront of self-supervised multi-camera depth estimation techniques. Through extensive evaluations and ablation studies, we have substantiated the contributions of each component of our framework, providing a comprehensive understanding of its strengths. As autonomous driving systems continue to evolve, our work presents valuable insights toward more accurate and robust depth estimation, crucial for the safety and reliability of such systems in diverse real-world scenarios.

### Broader Impact Statement

While our work on self-supervised multi-camera depth estimation, embodied in the STViT framework, represents a substantial stride in enhancing depth perception for applications like autonomous driving, it is imperative to consider potential repercussions and societal impacts. Firstly, as with any technology deployed in real-world scenarios, our model's performance may be influenced by specific conditions, such as extreme weather or low-light environments. Users should be aware that while our Adversarial Geometry Regularization Module (AGR) significantly improves robustness, there may still be limitations in adverse conditions that could affect the reliability of depth estimations. Additionally, given the nature of self-supervised learning and its reliance on diverse datasets, there is a need for vigilance regarding potential biases that could inadvertently impact the model's behavior in different scenarios. Furthermore, the application of such technologies in safety-critical domains, like autonomous driving, underscores the importance of continual monitoring and updates to ensure the model's adaptability to evolving real-world conditions. We emphasize the necessity for transparent and ethical deployment practices to mitigate any unforeseen consequences and ensure the responsible use of advanced depth estimation models in practical settings.

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
