# OpenReview forum: "STViT: Improving Self-supervised Multi-Camera Depth Estimation with Spatial-Temporal Context and Adversarial Geometry Regularization"
_TMLR — Rejected by TMLR_

### Review · Reviewer_eFWp · 2023-12-20

**Summary Of Contributions:**

This paper enhances self-supervised multi-camera depth estimation by utilizing cross-camera and cross-frame correlations, along with adversarial depth regularization. The paper primarily presents three contributions: 1) The Spatial-Temporal Transformer (STTrans) extracts depth features by leveraging local spatial connectivity and temporal information across timestamps and cameras using attention mechanisms. 2) A color correction approach (STPCC) is applied to RGB images for self-supervision loss calculation, mitigating the adverse effects of varying illumination conditions. 3) An Adversarial Geometry Regularization (AGR) module, based on Generative Adversarial Networks, is introduced to enhance its generalization across various environments.

**Audience:**

Yes

**Broader Impact Concerns:**

I think it's sufficient.

**Claims And Evidence:**

Yes

**Requested Changes:**

The author considers STPCC as a contribution, but there is not much description of how other methods address the illumination variance problem in the 'Related Work' section.

In the description of the STTrans method, the author mentions Multi-Scale Patch Embedding; however, there is no detailed explanation or reference to this module in the paper. As a result, I am unsure about the specific implementation of this module.

The definition of I_{t} in Formula (1) is not mentioned earlier in the text. Does the definition imply that only the images of the adjacent 3 frames are used as input? Also, why is STPCC not required for the input images here?

In the context of STTrans Ablation, why isn't the testing performed by gradually removing one module at a time starting from the full model? Additionally, the enabling or disabling of the 'Spatial-temporal photometric loss' in the ablation study is not very clear.

The AGR proposed in the paper claims to enhance the performance of the method under adverse conditions, such as rainy weather and nighttime driving. However, the results of the ablation study are not very pronounced. It might be beneficial to include visual comparative graphs for clarification.

**Strengths And Weaknesses:**

The proposed approach builds a novel pipeline for multi-camera self-supervised depth estimation combined with temporal context. It has several componets and each of them has a very clear objective. Generally , the methodology is sound.  The experimental results looks good and  demonstrates its effectiveness.  This paper also provides a systematic classification of existing methods, which is good.

However, there are still some issues in this article. Although the proposed methods are novel and reasonable, the improvement compared to the state-of-the-art results is not very noticeable. The experiments also fail to demonstrate whether the obtained improvement is due to an increase  in the number of network's parameters or a change in the method itself. There are unclear descriptions in the method description section, and I have some questions about the experimental part, as detailed in the 'Requested Changes'.

---

### Review · Reviewer_Eejt · 2024-01-22

**Summary Of Contributions:**

The authors propose a framework for the self-supervised training of models for multi-camera depth estimation.  Their method consists of three components: a Spatial-Temporal Transformer to exploit local connectivity and global context, a spatial-temporal photometric consistency correction to handle illumination variability, and a GAN-based regularizer to address adverse conditions such as rain.  Performance is on part with SOTA approaches EGA-Depth and MCDP.

**Audience:**

Yes

**Broader Impact Concerns:**

No concern

**Claims And Evidence:**

Yes

**Requested Changes:**

* Overall the performance is on par with EGA-Depth and occasionally MCDP.  Could any of these 3 components be added to those approach to produce yet a better model?  There needs to be additional discussion around what impact this work has, given its roughly similar performance to other approaches.  It's very clear what the authors have done, and what contribution to the final score each of these components have, but the contribution to the field is a little obvious.  Are these general approaches that can be used anywhere?  Is this a simpler or lighter-weight approach than others?  If it's simply an alternate approach to these others, I think that's ok, but it should be stated.  But if there's advantage or key difference between this approach and those, that needs to be made clearer.   The fact that this approach works and is well implemented is a positive, but the contribution to the field in its current form is not clear to me.

* Inclusion of error bars around the various results is important here given how small the difference is between different models.

* Please speak to performance and computational requirements both for this model as well as the other approaches.

* Please comment more on 4.5.1 where the performance of individual cameras vary.  Is the model trained to not be camera-ordering-invariant?  Or it is trained in such a manner, but at inference time it still struggles on back and right views?  Is this result seen for all approaches, or just the current approach?

* Can you speak to and/or quantify the impact of the window overlap amount?  It is mentioned that both datasets have very limited overlap.  If this overlap is increased or decreased synthetically (by cropping the images, for example), does that impact the model's performance?

* A visual example (and potentially residuals) from each SOTA approach examined might be useful in seeing the difference between these approaches to understand what errors are being made by each.

Most comments below are stylistic suggestions
<u> General </u>
The use of commas throughout is inconsistent.  In general, clauses beginning with "which" should have a comma, whereas clauses with "that" do not.
Some of the sentences, while properly written, are very long with multiple clauses.  It may be helpful to break these up into two sentences.  Often this occurs when describing what something is and what the impact is- these two items could be split into two sentences on occasion.

<u>Related Works</u>
Unsure why the end of 2.1 is in blue and bold.  In general this section is less clear than the others.  This section actually changes focus- to specifically on adverse conditions.  At a minimum it should probably be a new paragraph (with maybe come clarity to the language), or perhaps its own subsection.  Or it should be moved to 2.4?

Opening of 2.3 is not as clear as other sections.  The first sentence could be improved in its structure. Similarly the opening of 2.4 could be clarified as well- it does not read as smoothly as other sections and has a few typos. "One line of work... utilizes GAN-based <missing a noun> as".

It might be helpful to conclude at the end of each section how the current work builds on these past works.  So if it's possible to say something like "our method most closely resembles X", or "our method extends X by", just to help the reader further make that connection.  This is done in section 2.4 and is very clear and helpful.

2.4  "However, the former.... depth map <Add comma> while the later..."

<u>Method</u>
Opening of 3.1.1 could be tightened.  All of the sentences individually are clear, but the flow could be improved.  The first sentence is a general statement about SSL. The second sentence is largely transitional saying a 6 camera system has a lot of information.  How this translates to implications for local and global features is not strongly laid.  These three sentences are a bit weak and general- my suggestion would be to try to make the driving point more explicit.  It could even be "abrupt" like saying "We seek to design a method which more efficiently extracts local and global features across multiple cameras and timeframes...".  I think those three sentences could be combined to one and that would increase the clarity.  There is a lot of information in 3.1.1.  It might make sense to set it up as (even using bullet points)- Problem, past solution, proposed solution- and do that for each, visually breaking them out either as individual paragraphs or bullet points.

3.1.2 "importance of extracting effective features".  I'm not sure what this actually means.  Ineffective features are by definition not useful... What types of features do these papers suggest are important?  Are these contextual RGB features (e.g. that the object is a horse), or something else?

3.1.2 STTrans, "After that..."- check the plural/singular and use of articles in this sentence.

**Strengths And Weaknesses:**

<u>Strengths:</u>
The text is mostly clear throughout.  The introduction is very well written and sets the stage for what has been done in along this line of research previously and how the authors plan to build on it.  Even for someone who has not worked on this specific problem (multi-camera depth perception) previously, it is very straightforward to jump into the work and come quickly up to speed.

The approach is reasonable and straightforward.  The main idea around the proposed architecture is essentially to ensure connections between all spatiotemporal neighbors are explicitly mapped.

Results are compared to other SOTA methods and match the current leader (Shi et al 2023).

Ablation studies for both the overall framework as well as the  STTrans itself capture the impact of each proposed addition and suggest each is relevant and contributing to the overall performance.

<u>Weaknesses:</u>
The idea of leveraging overlapping views (spatially and/or temporally) is not particularly novel, nor has it been done in a particularly unique way in the proposed approach.   Use of intensity normalization and GAN-based corrections are also fairly standard common approaches.  However, these different approaches and techniques seem to be pulled together here in a clean manner.

It is important to see multiple runs to quantify the uncertainty around the results to determine which are statistically significant, especially given how similar some of the results are.

The results are roughly equivalent to EGA-Depth (Shi et al, 2023) and MCDP (Xu et al 2020a).  That is not a disqualifying result, only that it does not sizably move forward performance in this space.  The challenge is, without knowing other pieces of information like computational requirements or inference speed, why chose this approach over one of the others (as they appear statistically the same in many cases)?  Or are these three components independent enough that they could be picked up and used in with any framework?  Without this additional discussion it's hard to assess the impact this work would have.

Computational performance (e.g. training time, inference speed, memory requirements) is not discussed.  This is particularly important give the roughly similar performance to Shi et al, 2023.  If this is a more efficient approach, that has additional value.  Conversely, if this is much heavier and/or slower, what is the advantage over a lighter-weight method that is similarly performant?

---

> ### Author Response · Authors · 2024-02-18
> **Response to Reviewer  Eejt**
>
> Q1: Further discussion is needed to clarify its unique contributions to the field and whether it offers distinct advantages or simplicity compared to other methods.
>
> A1: As a newly proposed self-supervised multi-camera depth estimation algorithm, our method contributes as follows:
>
>         Regarding technical novelty, we propose the first transformer-based framework to investigate the self-supervised multi-camera depth estimation task. Specifically, we first propose a Spatial-Temporal Transformer (STTrans) for comprehensive feature extraction with further exploration of both cross-camera and cross-frame geometric correlations. As shown in Figure 2, STTrans exploits not only cross-camera correlation in the same frame (denoted by yellow arrows) and cross-frame correlation of the same camera (displayed as views with the corresponding same color) but also cross-camera and cross-frame correlation (different cameras in different frames, shown as colorful arrows among different temporal views) simultaneously. Previous methods SurroundDepth, MCDP, and EGA-Depth only exploit the cross-camera correlation in the same frame (denoted by yellow arrows).
>
>         Second, we introduce the Spatial-Temporal Photometric Consistency Correction (STPCC) strategy to enforce the brightness consistency of diverse views to improve the photometric loss calculation.
>
>         Third, we introduce an Adversarial Geometry Regularization (AGR) module to provide spatial positional restrictions for predicted depth maps, mitigating prediction weirdness in challenging cases.
>
>         These three modules are all newly proposed techniques to address the neglected issues in previous methods in the field of self-supervised multi-camera depth estimation. The detailed experiments further prove the effectiveness of these techniques.
>
>         Regarding experimental results, our method achieves state-of-the-art performance. On the main benchmark Nuscenes dataset, our method improves Abs Rel and RMSE by 2.5% and 1.7% respectively compared with EGA-Depth, and improves by 1.7% and 2% respectively compared with MCDP. Our method can produce better estimation results and we believe it can perform as a new state-of-the-art approach.
>
> Q2: Inclusion of error bars around the various results is important here given how small the difference is between different models.
>
> A2: Thanks for your valuable suggestions. We include error bars for our method and other methods, as illustrated in Table 1 and Table 2 in the revised version. As the source code of some methods (EGA-Depth and MCDP) has not been released, we can only conclude the results of our method and SurroundDepth, according to 5 times inference.
>
> Q3: Please speak to performance and computational requirements both for this model as well as the other approaches.
>
> A3: Thanks for your question. We compare the model efficiency in Table 11 in the revised version. According to the experiment results, our method can achieve a better performance without much computational burden. Notably, the improvement in Spatial-Temporal Photometric Consistency Correction and Adversarial Geometry Regularization Module are only applied during training, which has no impact on inference efficiency.
>
> Q4: Please comment more on 4.5.1 where the performance of individual cameras vary. Is the model trained to not be camera-ordering-invariant? Or it is trained in such a manner, but at inference time it still struggles on back and right views? Is this result seen for all approaches, or just the current approach?
>
> A4: The model is invariant to camera ordering during training, as long as the camera ordering between time series frames is kept consistent. However, the model still struggles with back and front views during inference. This experimental result can also be seen in other methods (SurroundDepth). We analyze that this phenomenon is caused by the different scenes captured by cameras from different orientations. For example, the camera on the right captures more dynamic object scenes than the left, making it more challenging. We report this experimental phenomenon and hope to provide valuable insights for subsequent research and improvement.

---

> ### Author Response · Authors · 2024-02-18
> **Response to Reviewer Eejt-2**
>
> Q5: Can you speak to and/or quantify the impact of the window overlap amount? It is mentioned that both datasets have very limited overlap. If this overlap is increased or decreased synthetically (by cropping the images, for example), does that impact the model's performance?
>
> A5: Thanks for your interesting question. We supplement an experiment here to show the impact of overlap on the model's performance by decreasing the overlap regions. We apply a mask to exclude different proportions of the overlap area. As shown in Figure 8 in the revised version, we illustrate the overlap region and the applied mask, taking $1/3$ masking in Front-Left, Front, and Front-right cameras as an example. According to the experiment results, model performance improves as the proportion of the overlap area increases, verifying the value of view overlaps.
>
> Q6: A visual example from each SOTA approach examined might be useful in seeing the difference between these approaches to understand what errors are being made by each.
>
> A6: Thanks for your valuable suggestions. We include a visual comparison in Figure 7 of our submission. Following your advice, we supplement more visual comparisons in the revised Figure 8. Because the source code of EGA-Depth and MCDP has not been released, we only list the results of our method and SurroundDepth.

---

### Review · Reviewer_6Sxt · 2024-02-05

**Summary Of Contributions:**

This paper works on multi-camera and multi-frame depth estimation in a self-supervised manner. The network structure adopts a spatial-temporal transformer structures that fuse both intra frame information across different cameras and inter frame information within the same camera. To capture the information from multiple scales, the authors apply convolution kernels with various sizes to a feature map, and concatenate the multi-scale feature maps from parallel transformer blocks along channel dimension before the spatial-temporal cross attention. To make the photometric loss more equivalent to various lighting changes, the authors adopt contrast limited histogram equalization to preprocess the input RGB images. In addition, the authors also use to a GAN based loss to enforce the spatial prior of depth maps. To demonstrate the effectiveness of the proposed method, the authors conduct experiments on NuScenes and DDAD and achieve superior performance over previous self-supervised multi-camera depth estimation methods.

**Audience:**

Yes

**Broader Impact Concerns:**

The authors have sufficiently discussed the broader impact of the proposed method.

**Claims And Evidence:**

Yes

**Requested Changes:**

In general, the paper is not ready to be published on TMLR  as it is.
I would suggest the authors to address the concerns listed in the weaknesses.

**Strengths And Weaknesses:**

Strengths:
+ The problem itself is well introduced.
+ Previous works are sufficiently reviewed and discussed.
+ A good combination of various existing network recipes.

Weaknesses:
- The paper lacks sufficient insights. The claimed contribution especially the STTrans and STPCC are weak.
  - STT: MPViT + additional cross attention (within a group of spatial-temporal frames with overlap).
  - STPCC: standard Histogram Equalization preprocessing.
- AGR lacks some important details. Specifically, whether the positional query is dependent on camera index $i$ is not introduced. Different camera has different spatial prior for depth maps, especially between the forward  and side cameras. Since the discriminator introduced in the paper is shared across cameras, I assume the positional query should be view-dependent. In addition, the structure of the discriminator should also be introduced. Because the generator and discriminator are expected to be symmetric as possible, and the generator in this paper is more than a standard vision transformer.
- The pose network seems unnecessary here. Since this paper is about depth estimation, instead of using a pose network and train with depth network jointly. It should be better to use "ground-truth" relative pose from (visual) odometry and focus on the depth estimation itself.

---

> ### Author Response · Authors · 2024-02-18
> **Response to Reviewer 6Sxt**
>
> Q1: The paper lacks sufficient insights. The claimed contribution especially the STTrans and STPCC are weak.
>
> A1: Thanks for your valuable comments.
>
>         Although the referenced MPViT and Histogram Equalization are existing techniques, we make modifications and adapt these techniques to our specific task of self-supervised multi-camera depth estimation. For STTrans, we are the first work to adopt transformer-based architecture for the self-supervised multi-camera depth estimation task with the first exploration of both cross-camera and cross-frame geometric correlations. Previous references including SurroundDepth, MCDP, and EGA-Depth only focus on the cross-camera correlations.
>
>         For STPCC, we modified Contrast Limited Histogram Equalization (CLHE) to correct image brightness and make the image color spatially and temporally consistent. This is very significant for the spatial-temporal photometric loss calculation, which is further helpful for improving depth prediction quality.
>
>         Last, we conduct extensive evaluations and studies of our method on large-scale autonomous driving datasets, demonstrating the effectiveness of the proposed modules. To sum up, our method achieves superior performance and advances the state-of-the-art in the multi-camera self-supervised depth estimation field.
>
> Q2: AGR lacks some important details. Specifically, whether the positional query is dependent on camera index is not introduced. Different camera has different spatial prior for depth maps, especially between the forward and side cameras. Since the discriminator introduced in the paper is shared across cameras, I assume the positional query should be view-dependent. In addition, the structure of the discriminator should also be introduced. Because the generator and discriminator are expected to be symmetric as possible, and the generator in this paper is more than a standard vision transformer.
>
> A2: Thanks for your valuable questions and suggestions. We supplement more details about AGR here. As shown in Figure 3, the input depth maps (predicted depth maps and arbitrary depth maps) for the discriminator should be listed in the same way according to the camera index to get the right regularization. Due to the different spatial prior in scenes captured by different cameras, the position query is dependent on the camera index. We adopt PatchGAN (a convolutional neural network-based discriminator in [1]) as the discriminator to make the prediction indistinguishable from the regularization reference. We agree that the generator and discriminator are expected to be as symmetric as possible in the random image generation task. However, PatchGAN can also be adapted with transformer-based encoders in other tasks, which is very common in previous works (please refer to Table IV in [2]). In our work, the depth generator is more important and requires spatio-temporal transformers to model 3D structures while the discriminator is just proposed to utilize the spatial positional priors to regularize the weirdness predictions in adverse conditions. We supplement the details of the discriminator in the revised version in green-color text.
>
> [1] Phillip Isola, Jun-Yan Zhu, Tinghui Zhou, and Alexei A Efros. Image-to-image translation with conditional adversarial networks. In Proceedings of the IEEE conference on computer vision and pattern recognition, pp. 1125–1134, 2017.
>
> [2] Dubey, Shiv Ram, and Singh, Satish Kumar. Transformer-based Generative Adversarial Networks in Computer Vision: A Comprehensive Survey. arXiv preprint arXiv:2302.08641,2023.
>
> Q3: The pose network seems unnecessary here. Since this paper is about depth estimation, instead of using a pose network and train with depth network jointly. It should be better to use "ground-truth" relative pose from (visual) odometry and focus on the depth estimation itself.
>
> A3: Thanks for your valuable comments. We focus on the self-supervised multi-camera depth estimation task in this work. The self-supervised paradigm requires the pose estimation simultaneously by utilizing the photometric loss. This is a general paradigm in this field, which has also been adopted in previous works including FSM, SurroudDepth, MCDP, EGA-Depth, etc.

---

### Decision · Action_Editor_ZxqW · 2024-03-29

**Recommendation:** Reject

**Comment:**

This work received two negative reviews and one relatively positive review. Its primary technical innovation involves the utilization of spatio-temporal transformers for self-supervised multi-camera depth estimation. However, this innovation appears to be relatively straightforward to develop. Furthermore, the work lacks substantial experimental evidence to establish the consistent superiority of the proposed approach over competing methods. It is helpful to supplement sufficient experimental evidence, e.g., more superior evaluation results on more benchmarks, to better demonstrate the proposed approach.

After considering both the strengths and weaknesses of the paper, the AE recommends rejection due to its inadequate form. Nonetheless, a resubmission with a major revision is encouraged.

**Audience:**

Yes

**Claims And Evidence:**

This work is deficient in experimental evidence that consistently demonstrates the superiority of the proposed approach over competing methods.

**Resubmission Of Major Revision:**

The authors may consider submitting a major revision at a later time.